# Stochastic Submodular Maximization:
# The Case of Coverage Functions

**Mohammad Reza Karimi**
Department of Computer Science
ETH Zurich
mkarimi@ethz.ch

**Mario Lucic**
Department of Computer Science
ETH Zurich
lucic@inf.ethz.ch

**Hamed Hassani**
Department of Electrical and Systems Engineering
University of Pennsylvania
hassani@seas.upenn.edu

**Andreas Krause**
Department of Computer Science
ETH Zurich
krausea@ethz.ch

## Abstract

Stochastic optimization of *continuous* objectives is at the heart of modern machine learning. However, many important problems are of *discrete* nature and often involve *submodular* objectives. We seek to unleash the power of stochastic continuous optimization, namely stochastic gradient descent and its variants, to such discrete problems. We first introduce the problem of *stochastic submodular optimization*, where one needs to optimize a submodular objective which is given as an expectation. Our model captures situations where the discrete objective arises as an empirical risk (e.g., in the case of exemplar-based clustering), or is given as an explicit stochastic model (e.g., in the case of influence maximization in social networks). By exploiting that common extensions *act linearly* on the class of submodular functions, we employ projected stochastic gradient ascent and its variants in the continuous domain, and perform rounding to obtain discrete solutions. We focus on the rich and widely used family of weighted coverage functions. We show that our approach yields solutions that are guaranteed to match the optimal approximation guarantees, while reducing the computational cost by several orders of magnitude, as we demonstrate empirically.

## 1 Introduction

Submodular functions are discrete analogs of convex functions. They arise naturally in many areas, such as the study of graphs, matroids, covering problems, and facility location problems. These functions are extensively studied in operations research and combinatorial optimization [22]. Recently, submodular functions have proven to be key concepts in other areas such as machine learning, algorithmic game theory, and social sciences. As such, they have been applied to a host of important problems such as modeling valuation functions in combinatorial auctions, feature and variable selection [23], data summarization [27], and influence maximization [20].

Classical results in submodular optimization consider the *oracle model* whereby the access to the optimization objective is provided through a black box — an oracle. However, in many applications, the objective has to be estimated from data and is subject to stochastic fluctuations. In other cases the value of the objective may only be obtained through simulation. As such, the exact computation might not be feasible due to statistical or computational constraints. As a concrete example, consider the problem of *influence maximization* in social networks [20]. The objective function is defined as the expectation of a stochastic process, quantifying the size of the (random) subset of nodes

influenced from a selected seed set. This expectation cannot be computed efficiently, and is typically approximated via random sampling, which introduces an error in the estimate of the value of a seed set. Another practical example is the *exemplar-based clustering* problem, which is an instance of the *facility location* problem. Here, the objective is the sum of similarities of all the points inside a (large) collection of data points to a selected set of centers. Given a distribution over point locations, the true objective is defined as the expected value w.r.t. this distribution, and can only be approximated as a sample average. Moreover, evaluating the function on a sample involves computation of many pairwise similarities, which is computationally prohibitive in the context of massive data sets.

In this work, we provide a formalization of such *stochastic submodular maximization tasks*. More precisely, we consider set functions $f : 2^V \to \mathbb{R}_+$, defined as $f(S) = \mathbb{E}_{\gamma \sim \Gamma}[f_\gamma(S)]$ for $S \subseteq V$, where $\Gamma$ is an arbitrary distribution and for each realization $\gamma \sim \Gamma$, the set function $f_\gamma : 2^V \to \mathbb{R}_+$ is monotone and submodular (hence $f$ is monotone submodular). The goal is to maximize $f$ subject to some constraints (e.g. the $k$-cardinality constraint) having access only to i.i.d. samples $f_{\gamma \sim \Gamma}(\cdot)$.

Methods for submodular maximization fall into two major categories: (i) The classic approach is to directly optimize the objective using discrete optimization methods (e.g. the GREEDY algorithm and its accelerated variants), which are state-of-the-art algorithms (both in practice and theory), at least in the case of simple constraints, and are most widely considered in the literature; (ii) The alternative is to lift the problem into a continuous domain and exploit continuous optimization techniques available therein [7]. While the continuous approaches may lead to provably good results, even for more complex constraints, their high computational complexity inhibits their practicality.

In this paper we demonstrate how modern stochastic optimization techniques (such as SGD, ADA-GRAD [8] and ADAM [21]), can be used to solve an important class of discrete optimization problems which can be modeled using weighted coverage functions. In particular, we show how to efficiently maximize them under matroid constraints by (i) lifting the problem into the continuous domain using the *multilinear extension* [37], (ii) efficiently computing a concave relaxation of the multilinear extension [32], (iii) efficiently computing an unbiased estimate of the gradient for the concave relaxation thus enabling (projected) stochastic gradient ascent-style algorithms to maximize the concave relaxation, and (iv) rounding the resulting fractional solution without loss of approximation quality [7]. In addition to providing convergence and approximation guarantees, we demonstrate that our algorithms enjoy strong empirical performance, often achieving an order of magnitude speedup with less than $1\%$ error with respect to GREEDY. As a result, the presented approach unleashes the powerful toolkit of stochastic gradient based approaches to discrete optimization problems.

**Our contributions.** In this paper we (i) introduce a framework for *stochastic submodular optimization*, (ii) provide a general methodology for constrained maximization of stochastic submodular objectives, (iii) prove that the proposed approach guarantees a $(1 - 1/e)-$approximation in expectation for the class of weighted coverage functions, which is the best approximation guarantee achievable in polynomial time unless P = NP, (iv) highlight the practical benefit and efficiency of using continuous-based stochastic optimization techniques for submodular maximization, (v) demonstrate the practical utility of the proposed framework in an extensive experimental evaluation. We show for the first time that continuous optimization is a highly practical, scalable avenue for maximizing submodular set functions.

## 2   Background and problem formulation

Let $V$ be a ground set of $n$ elements. A set function $f : 2^V \longrightarrow \mathbb{R}_+$ is *submodular* if for every $A, B \subseteq V$, it holds $f(A) + f(B) \geq f(A \cap B) + f(A \cup B)$. Function $f$ is said to be monotone if $f(A) \leq f(B)$ for all $A \subseteq B \subseteq V$. We focus on maximizing $f$ subject to some constraints on $S \subseteq V$. The prototypical example is maximization under the cardinality constraint, i.e., for a given integer $k$, find $S \subseteq V, |S| \leq k$, which maximizes $f$. Finding an exact solution for monotone submodular functions is NP-hard [10], but a $(1 - 1/e)$-approximation can be efficiently determined [30]. Going beyond the $(1 - 1/e)$-approximation is NP-hard for many classes of submodular functions [30, 24]. More generally, one may consider *matroid constraints*, whereby $(V, \mathcal{I})$ is a matroid with the family of independent sets $\mathcal{I}$, and maximize $f$ such that $S \in \mathcal{I}$. The GREEDY algorithm achieves a $1/2$-approximation [13], but CONTINUOUS GREEDY introduced by Vondrák [37], Calinescu et al. [6] can achieve a $(1 - 1/e)$-optimal solution in expectation. Their approach is based on the *multilinear*

*extension* of $f$, $F : [0, 1]^V \to \mathbb{R}_+$, defined as

$$F(\mathbf{x}) = \sum_{S \subseteq V} f(S) \prod_{i \in S} x_i \prod_{j \notin S} (1 - x_j), \tag{1}$$

for all $\mathbf{x} = (x_1, \cdots, x_n) \in [0, 1]^V$. In other words, $F(\mathbf{x})$ is the expected value of of $f$ over sets wherein each element $i$ is included with probability $x_i$ independently. Then, instead of optimizing $f(S)$ over $\mathcal{I}$, we can optimize $F$ over the matroid base polytope corresponding to $(V, \mathcal{I})$: $\mathcal{P} = \{\mathbf{x} \in \mathbb{R}_+^n \mid \mathbf{x}(S) \le r(S), \forall S \subseteq V, \mathbf{x}(V) = r(V)\}$, where $r(\cdot)$ is the matroid's rank function. The CONTINUOUS GREEDY algorithm then finds a solution $\mathbf{x} \in \mathcal{P}$ which provides a $(1 - 1/e)-$approximation. Finally, the continuous solution $\mathbf{x}$ is then efficiently rounded to a feasible discrete solution without loss in objective value, using PIPAGE ROUNDING [1, 6]. The idea of converting a discrete optimization problem into a continuous one was first exploited by Lovász [28] in the context of submodular minimization and this approach was recently applied to a variety of problems [36, 19, 3].

**Problem formulation.** The aforementioned results are based on the *oracle model*, whereby the exact value of $f(S)$ for any $S \subseteq V$ is given by an oracle. In absence of such an oracle, we face the additional challenges of *evaluating* $f$, both statistical and computational. In particular, consider set functions that are defined as *expectations*, i.e. for $S \subseteq V$ we have

$$f(S) = \mathbb{E}_{\gamma \sim \Gamma}[f_\gamma(S)], \tag{2}$$

where $\Gamma$ is an arbitrary distribution and for each realization $\gamma \sim \Gamma$, the set function $f_\gamma : 2^V \to \mathbb{R}$ is submodular. The goal is to efficiently maximize $f$ subject to constraints such as the $k$-cardinality constraint, or more generally, a matroid constraint.

As a motivating example, consider the problem of propagation of contagions through a network. The objective is to identify the most influential seed set of a given size. A propagation instance (concrete realization of a contagion) is specified by a graph $G = (V, E)$. The influence $f_G(S)$ of a set of nodes $S$ in instance $G$ is the fraction of nodes reachable from $S$ using the edges $E$. To handle uncertainties in the concrete realization, it is natural to introduce a probabilistic model such as the Independent Cascade [20] model which defines a distribution $\mathcal{G}$ over instances $G \sim \mathcal{G}$ that share a set $V$ of nodes. The influence of a seed set $S$ is then the expectation $f(S) = \mathbb{E}_{G \sim \mathcal{G}}[f_G(S)]$, which is a monotone submodular function. Hence, estimating the expected influence is computationally demanding, as it requires summing over exponentially many functions $f_G$. Assuming $f$ as in (2), one can easily obtain an unbiased estimate of $f$ for a fixed set $S$ by random sampling according to $\Gamma$. The critical question is, given that the underlying function *is* an expectation, can we optimize it more efficiently?

Our approach is based on continuous extensions that are linear operators on the class of set functions, namely, *linear continuous extensions*. As a specific example, considering the multilinear extension, we can write $F(\mathbf{x}) = \mathbb{E}_{\gamma \sim \Gamma}[F_\gamma(\mathbf{x})]$, where $F_\gamma$ denotes the extension of $f_\gamma$. As a consequence, the value of $F_\gamma(\mathbf{x})$, when $\gamma \sim \Gamma$, is an *unbiased estimator* for $F(\mathbf{x})$ and unbiased estimates of the (sub)gradients may be obtained analogously. We explore this avenue to develop efficient algorithms for maximizing an important subclass of submodular functions that can be expressed as weighted coverage functions. Our approach harnesses a *concave relaxation* detailed in Section 3.

**Further related work.** The emergence of new applications, combined with a massive increase in the amount of data has created a demand for fast algorithms for submodular optimization. A variety of approximation algorithms have been presented, ranging from submodular maximization subject to a cardinality constraint [29, 39, 4], submodular maximization subject to a matroid constraint [6], non-monotone submodular maximization [11], approximately submodular functions [17], and algorithms for submodular maximization subject to a wide variety of constraints [25, 12, 38, 18, 9]. A closely related setting to ours is online submodular maximization [35], where functions come one at a time and the goal is to provide time-dependent solutions (sets) such that a cumulative regret is minimized. In contrast, our goal is to find a single (time-independent) set that maximizes the objective (2). Another relevant setting is noisy submodular maximization, where the evaluations returned by the oracle are noisy [16, 34]. Specifically, [34] assumes a noisy but unbiased oracle (with an independent sub-Gaussian noise) which allows one to sufficiently estimate the marginal gains of items by averaging. In the context of cardinality constraints, some of these ideas can be carried to our setting by introducing additional assumptions on how the values $f_\gamma(S)$ vary w.r.t. to their expectation $f(S)$. However, we provide a different approach that does not rely on uniform convergence and compare sample and running time complexity comparison with variants of GREEDY in Section 3.

# 3 Stochastic Submodular Optimization

We follow the general framework of [37] whereby the problem is lifted into the continuous domain, a continuous optimization algorithm is designed to maximize the transferred objective, and the resulting solution is rounded. Maximizing $f$ subject to a matroid constraint can then be done by first maximizing its multilinear extension $F$ over the matroid base polytope and then rounding the solution. Methods such as the projected stochastic gradient ascent can be used to maximize $F$ over this polytope.

Critically, we have to assure that the computed local optima are *good* in expectation. Unfortunately, the multilinear extension $F$ lacks concavity and therefore may have bad local optima. Hence, we consider *concave* continuous extensions of $F$ that are *efficiently computable*, and at most a constant factor away from $F$ to ensure solution quality. As a result, such a concave extension $\bar{F}$ could then be efficiently maximized over a polytope using *projected stochastic gradient ascent* which would enable the application of modern continuous optimization techniques. One class of important functions for which such an extension can be efficiently computed is the class of weighted coverage functions.

**The class of weighted coverage functions (WCF).** Let $U$ be a set and let $g$ be a nonnegative modular function on $U$, i.e. $g(S) = \sum_{u \in S} w(u)$, $S \subseteq U$. Let $V = \{B_1, \ldots, B_n\}$ be a collection of subsets of $U$. The *weighted coverage function* $f : 2^V \longrightarrow \mathbb{R}^+$ defined as

$$\forall S \subseteq V : \; f(S) = g\left(\bigcup_{B_i \in S} B_i\right)$$

is monotone submodular. For all $u \in U$, let us denote by $P_u := \{B_i \in V \mid u \in B_i\}$ and by $\mathbb{I}(\cdot)$ the indicator function. The multilinear extension of $f$ can be expressed in a more compact way:

$$F(\mathbf{x}) = \mathbb{E}_S[f(S)] = \mathbb{E}_S \sum_{u \in U} \mathbb{I}(u \in B_i \text{ for some } B_i \in S) \cdot w(u)$$

$$= \sum_{u \in U} w(u) \cdot \mathbb{P}(u \in B_i \text{ for some } B_i \in S) = \sum_{u \in U} w(u)\left(1 - \prod_{B_i \in P_u}(1 - x_i)\right) \quad (3)$$

where we used the fact that each element $B_i \in V$ was chosen with probability $x_i$.

**Concave upper bound for weighted coverage functions.** To efficiently compute a concave upper bound on the multilinear extension we use the framework of Seeman and Singer [32]. Given that all the weights $w(u)$, $u \in U$ in (3) are non-negative, we can construct a concave upper bound for the multilinear extension $F(\mathbf{x})$ using the following Lemma. Proofs can be found in the Appendix A.

**Lemma 1.** *For* $\mathbf{x} \in [0, 1]^\ell$ *define* $\alpha(\mathbf{x}) := 1 - \prod_{i=1}^\ell (1 - x_i)$. *Then the Fenchel concave biconjugate of* $\alpha(\cdot)$ *is* $\beta(\mathbf{x}) := \min\left\{1, \sum_{i=1}^\ell x_i\right\}$. *Also*

$$(1 - 1/e)\,\beta(\mathbf{x}) \leq \alpha(\mathbf{x}) \leq \beta(\mathbf{x}) \quad \forall \mathbf{x} \in [0, 1]^\ell.$$

*Furthermore, $\beta$ is an extension of $\alpha$, i.e. $\forall \mathbf{x} \in \{0, 1\}^\ell$: $\alpha(\mathbf{x}) = \beta(\mathbf{x})$.*

Consequently, given a weighted coverage function $f$ with $F(\mathbf{x})$ represented as in (3), we can define

$$\bar{F}(\mathbf{x}) := \sum_{u \in U} w(u) \min\left\{1, \sum_{B_v \in P_u} x_v\right\} \quad (4)$$

and conclude using Lemma 1 that $(1 - 1/e)\bar{F}(\mathbf{x}) \leq F(\mathbf{x}) \leq \bar{F}(\mathbf{x})$, as desired. Furthermore, $\bar{F}$ has three interesting properties: (1) It is a concave function over $[0, 1]^V$, (2) it is equal to $f$ on vertices of the hypercube, i.e. for $\mathbf{x} \in \{0, 1\}^n$ one has $\bar{F}(\mathbf{x}) = f(\{i : x_i = 1\})$, and (3) it can be computed efficiently and deterministically given access to the sets $P_u$, $u \in U$. In other words, we can compute the value of $\bar{F}(\mathbf{x})$ using at most $\mathcal{O}(|U| \times |V|)$ operations. Note that $\bar{F}$ is not the *tightest* concave upper bound of $F$, even though we use the tightest concave upper bounds for each term of $F$.

**Optimizing the concave upper bound by stochastic gradient ascent.** Instead of maximizing $F$ over a polytope $\mathcal{P}$, one can now attempt to maximize $\bar{F}$ over $\mathcal{P}$. Critically, this task can be done efficiently, as $\bar{F}$ is concave, by using projected stochastic gradient ascent. In particular, one can

---

**Algorithm 1** Stochastic Submodular Maximization via concave relaxation

---

**Require:** matroid $\mathcal{M}$ with base polytope $\mathcal{P}$, $\eta_t$ (step size), $T$ (maximum # of iterations)

1: $\mathbf{x}^{(0)} \leftarrow$ starting point in $\mathcal{P}$
2: **for** $t \leftarrow 0$ **to** $T - 1$ **do**
3:     Choose $\mathbf{g}_t$ at random from a distribution such that $\mathbb{E}[\mathbf{g}_t | \mathbf{x}^{(0)}, \ldots, \mathbf{x}^{(t)}] \in \partial \bar{F}(\mathbf{x}^{(t)})$
4:     $\mathbf{x}^{(t+1/2)} \leftarrow \mathbf{x}^{(t)} + \eta_t \, \mathbf{g}_t$
5:     $\mathbf{x}^{(t+1)} \leftarrow \text{Project}_{\mathcal{P}}(\mathbf{x}^{(t+1/2)})$
6: **end for**
7: $\bar{\mathbf{x}}_T \leftarrow \frac{1}{T} \sum_{t=1}^{T} \mathbf{x}^{(t)}$
8: $S \leftarrow \text{RANDOMIZED-PIPAGE-ROUND}(\bar{\mathbf{x}}_T)$
9: **return** $S$ such that $S \in \mathcal{M}$, $\mathbb{E}[f(S)] \geq (1 - 1/e)f(OPT) - \varepsilon(T)$.

---

control the convergence speed by choosing from the toolbox of modern continuous optimization algorithms, such as SGD, ADAGRAD and ADAM. Let us denote a maximizer of $\bar{F}$ over $\mathcal{P}$ by $\bar{\mathbf{x}}^*$, and also a maximizer of $F$ over $\mathcal{P}$ by $\mathbf{x}^*$. We can thus write

$$F(\bar{\mathbf{x}}^*) \geq (1 - 1/e)\bar{F}(\bar{\mathbf{x}}^*) \geq (1 - 1/e)\bar{F}(\mathbf{x}^*) \geq (1 - 1/e)F(\mathbf{x}^*),$$

which is the exact guarantee that previous methods give, and in general is the best near-optimality ratio that one can give in poly-time. Finally, to round the continuous solution we may apply RANDOMIZED-PIPAGE-ROUNDING [7] as the quality of the approximation is preserved in expectation.

**Matroid constraints.** Constrained optimization can be efficiently performed by projected gradient ascent whereby after each step of the stochastic ascent, we need to project the solution back onto the feasible set. For the case of matroid constraints, it is sufficient to consider projection onto the matroid base polytope. This problem of projecting on the base polytope has been widely studied and fast algorithms exist in many cases [2, 5, 31]. While these projection algorithms were used as a key subprocedure in constrained submodular minimization, here we consider them for submodular maximization. Details of a fast projection algorithm for the problems considered in this work are presented the Appendix D. Algorithm 1 summarizes all steps required to maximize $f$ subject to matroid constraints.

**Convergence rate.** Since we are maximizing a concave function $\bar{F}(\cdot)$ over a matroid base polytope $\mathcal{P}$, convergence rate (and hence running time) depends on $B := \max_{\mathbf{x} \in \mathcal{P}} ||\mathbf{x}||$, as well as maximum gradient norm $\rho$ (i.e. $||\mathbf{g}_t|| \leq \rho$ with probability 1). [1] In the case of the base polytope for a matroid of rank $r$, $B$ is $\sqrt{r}$, since each vertex of the polytope has exactly $r$ ones. Also, from (4), one can build a rough upper bound for the norm of the gradient:

$$||\mathbf{g}|| \leq || \textstyle\sum_{u \in U} w(u) \mathbf{1}_{P_u}|| \leq \big(\max_{u \in U}|P_u|\big)^{1/2} \sum_{u \in U} w(u),$$

which depends on the weights $w(u)$ as well as $|P_u|$ and is hence problem-dependent. We will provide tighter upper bounds for gradient norm in our specific examples in the later sections. With $\eta_t = B/\rho\sqrt{t}$, and classic results for SGD [33], we have that

$$\bar{F}(\mathbf{x}^*) - \mathbb{E}[\bar{F}(\bar{\mathbf{x}}_T)] \leq B\rho/\sqrt{T},$$

where $T$ is the total number of SGD iterations and $\bar{\mathbf{x}}_T$ is the final outcome of SGD (see Algorithm 1). Therefore, for a given $\varepsilon > 0$, after $T \geq B^2\rho^2/\varepsilon^2$ iterations, we have

$$\bar{F}(\mathbf{x}^*) - \mathbb{E}[\bar{F}(\bar{\mathbf{x}}_T)] \leq \varepsilon.$$

Summing up, we will have the following theorem:

**Theorem 2.** *Let $f$ be a weighted coverage function, $\mathcal{P}$ be the base polytope of a matroid $\mathcal{M}$, and $\rho$ and $B$ be as above. Then for each $\epsilon > 0$, Algorithm 1 after $T = B^2\rho^2/\varepsilon^2$ iterations, produces a set $S^* \in \mathcal{M}$ such that $\mathbb{E}[f(S^*)] \geq (1 - 1/e) \max_{S \in \mathcal{M}} f(S) - \varepsilon$.*

**Remark.** Indeed this approximation ratio is the best ratio one can achieve, unless P=NP [10]. A key point to make here is that our approach also works for more general constraints (in particular is efficient for *simple* matroids such as partition matroids). In the latter case, GREEDY only gives $\frac{1}{2}$-approximation and fast discrete methods like STOCHASTIC-GREEDY [29] do not apply, whereas our method still yields an $(1 - 1/e)$-optimal solution.

**Time Complexity.** One can compute an upper bound for the running time of Algorithm 1 by estimating the time required to perform gradient computations, projection on $\mathcal{P}$, and rounding. For the case of uniform matroids, projection and rounding take $\mathcal{O}(n \log n)$ and $\mathcal{O}(n)$ time, respectively (see Appendix D). Furthermore, for the applications considered in this work, namely expected influence maximization and exemplar-based clustering, we provide linear time algorithms to compute the gradients. Also when our matroid is the $k$-uniform matroid (i.e. $k$-cardinality constraint), we have $B = \sqrt{k}$. By Theorem 2, the total computational complexity of our algorithm is $\mathcal{O}(\rho^2 kn(\log n)/\varepsilon^2)$.

**Comparison to GREEDY.** Let us relate our results to the classical approach. When running the GREEDY algorithm in the stochastic setting, one estimates $\hat{f}(S) := \frac{1}{s} \sum_{i=1}^{s} f_{\gamma_i}(S)$ where $\gamma_1, \ldots, \gamma_s$ are i.i.d. samples from $\Gamma$. The following proposition bounds the sample and computational complexity of GREEDY. The proof is detailed in the Appendix B.

**Proposition 3.** *Let $f$ be a submodular function defined as* (2). *Suppose $0 \leq f_\gamma(S) \leq H$ for all $S \subseteq V$ and all $\gamma \sim \Gamma$. Assume $S^*$ denotes the optimal solution for $f$ subject to $k$-cardinality constraint and $S_k$ denotes the solution computed by the greedy algorithm on $\hat{f}$ after $k$ steps. Then, in order to guarantee*

$$\mathbb{P}[f(S_k) \geq (1 - 1/e)f(S^*) - \varepsilon] \geq 1 - \delta,$$

*it is enough to have*

$$s \in \Omega\left(H^2(k \log n + \log(1/\delta))/\varepsilon^2\right),$$

*i.i.d. samples from $\Gamma$. The running time of GREEDY is then bounded by*

$$\mathcal{O}\left(\tau H^2 nk(k \log n + \log(1/\delta))/\varepsilon^2\right),$$

*where $\tau$ is an upper bound on the computation time for a single evaluation of $f_\gamma(S)$.*

As an example, let us compare the worst-case complexity bound obtained for SGD (i.e. $\mathcal{O}(\rho^2 kn(\log n)/\varepsilon^2)$) with that of GREEDY for the influence maximization problem. Each single function evaluation for GREEDY amounts to computing the total influence of a set in a sample graph, which makes $\tau = O(n)$ (here we assume our sample graphs satisfy $|E| = O(|V|)$). Also, a crude upper bound for the size of the gradient for each sample function is $H\sqrt{n}$ (see Appendix E.1). Hence, we can deduce that SGD can have a factor $k$ speedup w.r.t. to GREEDY.

## 4 Applications

We will now show how to instantiate the *stochastic submodular maximization framework* using several prototypical discrete optimization problems.

**Influence maximization.** As discussed in Section 2, the Independent Cascade [20] model defines a distribution $\mathcal{G}$ over instances $G \sim \mathcal{G}$ that share a set $V$ of nodes. The influence $f_G(S)$ of a set of nodes $S$ in instance $G$ is the fraction of nodes reachable from $S$ using the edges $E(G)$. The following Lemma shows that the influence belongs to the class of WCF.

**Lemma 4.** *The influence function $f_G(\cdot)$ is a WCF. Moreover,*

$$F_G(\mathbf{x}) = \mathbb{E}_S[f_G(S)] = \frac{1}{|V|} \sum_{v \in V} (1 - \prod_{u \in P_v}(1 - x_u)) \tag{5}$$

$$\bar{F}_G(\mathbf{x}) = \frac{1}{|V|} \sum_{v \in V} \min\{1, \sum_{u \in P_v} x_u\}, \tag{6}$$

*where $P_v$ is the set of all nodes having a (directed) path to $v$.*

We return to the problem of maximizing $f_{\mathcal{G}}(S) = \mathbb{E}_{G \sim \mathcal{G}}[f_G(S)]$ given a distribution over graphs $\mathcal{G}$ sharing nodes $V$. Since $f_{\mathcal{G}}$ is a weighted sum of submodular functions, it is submodular. Moreover,

$$F(\mathbf{x}) = \mathbb{E}_S[f_{\mathcal{G}}(S)] = \mathbb{E}_S[\mathbb{E}_G[f_G(S)]] = \mathbb{E}_G[\mathbb{E}_S[f_G(S)]] = \mathbb{E}_G[F_G(\mathbf{x})]$$

$$= \mathbb{E}_G\left[\frac{1}{|V|}\sum_{v \in V}(1 - \prod_{u \in P_v}(1 - x_u))\right].$$

Let $\mathcal{U}$ be the uniform distribution over vertices. Then,

$$F(\mathbf{x}) = \mathbb{E}_G\left[\frac{1}{|V|}\sum_{v \in V}(1 - \prod_{u \in P_v}(1 - x_u))\right] = \mathbb{E}_G\left[\mathbb{E}_{v \sim \mathcal{U}}\left[1 - \prod_{u \in P_v}(1 - x_u)\right]\right], \quad (7)$$

and the corresponding upper bound would be

$$\bar{F}(\mathbf{x}) = \mathbb{E}_G\left[\mathbb{E}_{v \sim \mathcal{U}}\left[\min\{1, \sum_{u \in P_v} x_u\}\right]\right]. \quad (8)$$

This formulation proves to be helpful in *efficient* calculation of subgradients, as one can obtain a random subgradient in linear time. For more details see Appendix E.1. We also provide a more efficient, *biased* estimator of the expectation in the Appendix.

**Facility location.** Let $G = (X \dot\cup Y, E)$ be a complete weighted bipartite graph with parts $X$ and $Y$ and nonnegative weights $w_{x,y}$. The weights can be considered as utilities or some similarity metric. We select a subset $S \subseteq X$ and each $y \in Y$ selects $s \in S$ with the highest weight $w_{s,y}$. Our goal is to maximize the average weight of these selected edges, i.e. to maximize

$$f(S) = \frac{1}{|Y|}\sum_{y \in Y}\max_{s \in S} w_{s,y} \quad (9)$$

given some constraints on $S$. This problem is indeed the *Facility Location* problem, if one takes $X$ to be the set of facilities and $Y$ to be the set of customers and $w_{x,y}$ to be the utility of facility $x$ for customer $y$. Another interesting instance is the *Exemplar-based Clustering* problem, in which $X = Y$ is a set of objects and $w_{x,y}$ is the similarity (or inverted distance) between objects $x$ and $y$, and one tries to find a subset $S$ of exemplars (i.e. *centroids*) for these objects.

The stochastic nature of this problem is revealed when one writes (9) as the expectation $f(S) = \mathbb{E}_{y \sim \Gamma}[f_y(S)]$, where $\Gamma$ is the uniform distribution over $Y$ and $f_y(S) := \max_{s \in S} w_{s,y}$. One can also consider this more general case, where $y$'s are drawn from an unknown distribution, and one tries to maximize the aforementioned expectation.

First, we claim that $f_y(\cdot)$ for each $y \in Y$ is again a weighted coverage function. For simplicity, let $X = \{1, \ldots, n\}$ and set $m_i \doteq w_{i,y}$, with $m_1 \geq \cdots \geq m_n$ and $m_{n+1} \doteq 0$.

**Lemma 5.** *The utility function $f_y(\cdot)$ is a WCF. Moreover,*

$$F_y(\mathbf{x}) = \sum_{i=1}^n (m_i - m_{i+1})(1 - \prod_{j=1}^i (1 - x_j)), \quad (10)$$

$$\bar{F}_y(\mathbf{x}) = \sum_{i=1}^n (m_i - m_{i+1})\min\{1, \sum_{j=1}^i x_j\}. \quad (11)$$

We remark that the gradient of both $F_y$ and $\bar{F}_y$ can be computed in linear time using a recursive procedure. We refer to Appendix E.2 for more details.

## 5 Experimental Results

We demonstrate the practical utility of the proposed framework and compare it to standard baselines. We compare the performance of the algorithms in terms of their wall-clock running time and the obtained utility. We consider the following problems:

- **Influence Maximization for the Epinions network[2].** The network consists of 75 879 nodes and 508 837 directed edges. We consider the subgraph induced by the top 10 000 nodes with the largest out-degree and use the independent cascade model [20]. The diffusion model is specified by a fixed probability for each node to influence its neighbors in the underlying graph. We set this probability $p$ to be 0.02, and chose the number of seeds $k = 50$.

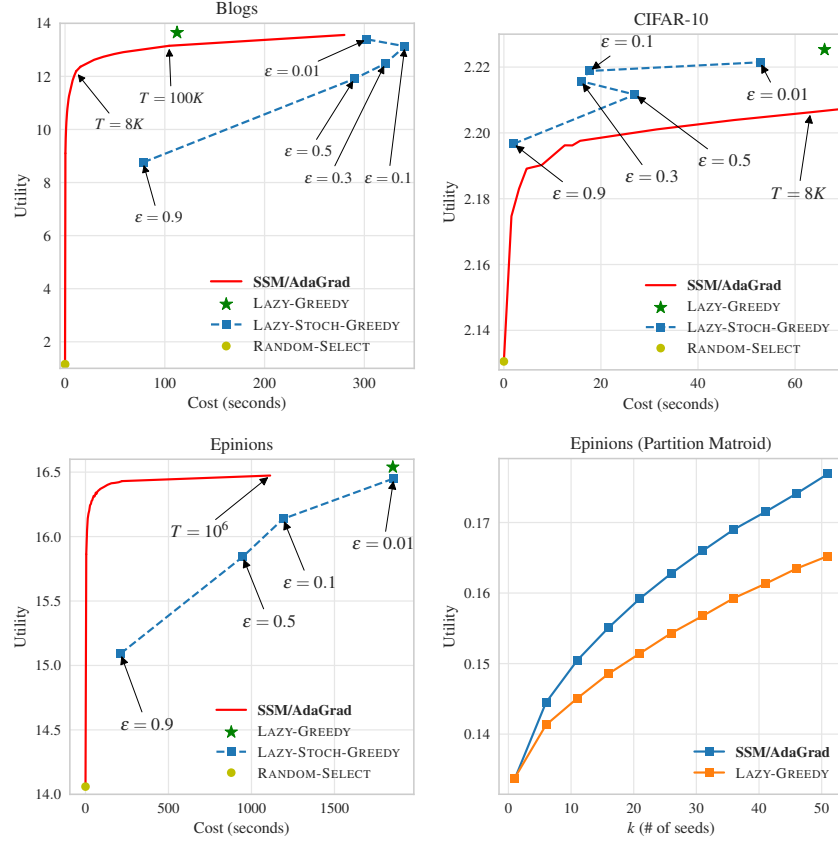

Figure 1: In the case of Facility location for Blog selection as well as on influence maximization on Epinions, the proposed approach reaches the same utility *significantly* faster. On the exemplar-based clustering of CIFAR, the proposed approach is outperformed by STOCHASTIC-GREEDY, but nevertheless reaches $98.4\%$ of the GREEDY utility in a few seconds (after less than 1000 iterations). On Influence Maximization over *partition matroids*, the proposed approach significantly outperforms GREEDY.

- **Facility Location for Blog Selection.** We use the data set used in [14], consisting of $45\,193$ blogs, and $16\,551$ cascades. The goal is to detect information cascades/stories spreading over the blogosphere. This dataset is *heavy-tailed*, hence a small random sample of the events has high variance in terms of the cascade sizes. We set $k = 100$.

- **Exemplar-based Clustering on CIFAR-10.** The data set contains $60\,000$ color images with resolution $32 \times 32$. We use a single batch of $10\,000$ images and compare our algorithms to variants of GREEDY over the full data set. We use the Euclidean norm as the distance function and set $k = 50$. Further details about preprocessing of the data as well as formulation of the submodular function can be found in Appendix E.3.

**Baselines.** In the case of cardinality constraints, we compare our stochastic continuous optimization approach against the most efficient discrete approaches (LAZY-)GREEDY and (LAZY-)STOCHASTIC-GREEDY, which both provide optimal approximation guarantees. For STOCHASTIC-GREEDY, we vary the parameter $\varepsilon$ in order to explore the running time/utility tradeoff. We also report the performance of randomly selected sets. For the two facility location problems, when applying the greedy variants we can evaluate the exact objective (true expectation). In the Influence Maximization application, computing the exact expectation is intractable. Hence, we use an empirical average of $s$ samples (cascades) from the model. We note that the number of samples suggested by Proposition 3 is overly conservative, and instead we make a practical choice of $s = 10^3$ samples.

**Results.** The results are summarized in Figure 1. On the blog selection and influence maximization applications, the proposed continuous optimization approach outperforms STOCHASTIC-GREEDY in terms of the running time/utility tradeoff. In particular, for blog selection we can compute a solution with the same utility $26\times$ faster than STOCHASTIC-GREEDY with $\varepsilon = 0.5$. Similarly, for influence maximization on Epinions we the solution $88\times$ faster than STOCHASTIC-GREEDY with $\varepsilon = 0.1$. On the exemplar-based clustering application STOCHASTIC-GREEDY outperforms the proposed approach. We note that the proposed approach is still competitive as it recovers $98.4\%$ of the value after less than thousand iterations.

We also include an experiment on Influence Maximization over *partition matroids* for the Epinions network. In this case, GREEDY only provides a $1/2$ approximation guarantee and STOCHASTIC-GREEDY does not apply. To create the partition, we first sorted all the vertices by their out-degree. Using this order on the vertices, we divided the vertices into two partitions, one containing vertices with even positions, other containing the rest. Figure 1 clearly demonstrates that the proposed approach outperforms GREEDY in terms of utility (as well as running time).

**Acknowledgments** The research was partially supported by ERC StG 307036. We would like to thank Yaron Singer for helpful comments and suggestions.

## Footnotes

[1]Note that the function $\bar{F}$ is neither smooth nor strongly concave as functions such as $\min\{1, x\}$ are not smooth or strongly concave.

[2]http://snap.stanford.edu/

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
