[Supplementary Material]

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

# A Proof of Lemma 1

Here we prove the inequality mentioned in the lemma. Proof of the fact of being Fenchel biconjugate is in Appendix F.

We prove the left-hand-side inequality, since the right-hand-side inequality is a consequence of Fenchel biconjugate-ness.

Let $\theta := \sum_{i=1}^{\ell} x_i$. We note from the inequality $1 - x \leq \exp(-x)$ that $\prod_{i=1}^{\ell}(1 - x_i) \leq \exp(-\theta)$. We thus obtain

$$1 - \prod_{i=1}^{\ell}(1 - x_i) \geq 1 - \exp(-\theta).$$

Now, if $\theta \geq 1$ then the result is clear. Also, if $\theta < 1$, then we note that the function $(1 - \exp(-\theta))/\theta$ is decreasing for $\theta \in (0,1)$, and hence, $1 - \exp(-\theta) \geq \theta(1 - 1/e)$. The left-hand-side inequality thus follows immediately.

# B Proof of Proposition 3

Note that the total number of subsets of cardinality less than $k$ is bounded from above by $k\binom{n}{k}$. For each such set $S$ we want the estimate $\hat{f}(S) := \frac{1}{s}\sum_{i=1}^{s} f_{\gamma_i}(S)$ to be at most $\epsilon$ away from $f(S)$. Also, note that the function $\hat{f}$ is itself a submodular function and maximizing it would give a $(1 - 1/e)$-approximation to its optimum. Hence, it is enough to have enough samples such that for all subsets $S$ of cardinality at most $k$ the two values $f(S)$ and $\hat{f}(S)$ differ by at most epsilon. By using Hoeffding's inequality and a union bound over all the subsets of cardinality at most $k$ (note that $\log(n\binom{n}{k}) = \mathcal{O}(k\log(n)))$ we get the result.

# C Proof of Lemmas 4 and 5

## C.1 Lemma 4

*Proof.* Let $A := \{C_v \mid v \in V\}$, where $C_v$ is the set of vertices reachable from $v$. By construction, there is a one-to-one correspondence between elements of $A$ and $V$, namely $C_v \leftrightarrow v$. For $T \subseteq A$, let $S \subseteq V$ be its corresponding subset in $V$, i.e. $S = \{v \in V \mid v \leftrightarrow C_v, C_v \in T\}$. It's obvious that $\bigcup_{v \in S} C_v = \bigcup_{C_v \in T} C_v$. Setting $g(T) = \frac{|T|}{|V|}$, makes $f'_G(T) := g(\bigcup_{C_v \in T} C_v)$ a WCF. But $f'_G(T) = f_G(S)$, so $f_G(\cdot)$ is also a WCF.

Moreover, for each $v \in V$, the set $P_v$ is the set of all elements of $A$ that contain $v$, which are precisely those vertices from which there is a (directed) path to $v$. We also relax our notation, and replace any element of $A$ by its correspondent in $V$. Hence,

$$F_G(\mathbf{x}) = \mathbb{E}_S[f_G(S)] = \frac{1}{|V|} \sum_{v \in V}(1 - \prod_{u \in P_v}(1 - x_u))$$

$$\bar{F}_G(\mathbf{x}) = \frac{1}{|V|} \sum_{v \in V} \min\{1, \sum_{u \in P_v} x_u\},$$

which are poly-time computable since one can find $P_v$ with a simple BFS algorithm in $\mathcal{O}(|V| + |E|)$ for each $v \in V$.

$\square$

## C.2 Lemma 5

*Proof.* Write $f(\cdot)$ instead of $f_y(\cdot)$.

Let $V = \{C_i \mid 1 \leq i \leq n\}$, where $C_i = \{i, \ldots, n\}$, and let $w(i) = m_i - m_{i+1}$ (set $m_{n+1} = 0$). Note that there is a natural bijection between $V$ and $U$, namely $C_i \leftrightarrow i$. Let $g$ be the modular function with weights $w(i)$, defined on $2^U$, and define the WCF $f' : 2^V \to \mathbb{R}_+$ as

$$f'(S) := g(\bigcup_{i \in S} C_i) = \sum_{j \in \bigcup_{i \in S} C_i} w(j). \tag{12}$$

Since $C_i$'s are forming a decreasing chain, $\bigcup_{i \in S} C_i = C_{\min S}$ and (12) becomes

$$f'(S) = \sum_{j \in C_{\min S}} w(j) = \sum_{j=\min S}^{n} w(j) = m_{\min S} - m_{n+1} = \max_{i \in S} m_i,$$

which is exactly $f(S)$.

Furthermore, $P_i$ is simply the set $\{1, \ldots, i\}$. Hence, we can write the multilinear extension and the corresponding upper bound as

$$F_y(\mathbf{x}) = \sum_{i=1}^{n} (m_i - m_{i+1})(1 - \prod_{j=1}^{i}(1 - x_j)),$$
$$\bar{F}_y(\mathbf{x}) = \sum_{i=1}^{n} (m_i - m_{i+1}) \min\{1, \sum_{j=1}^{i} x_j\}.$$

$\square$

# D   Fast Algorithms for Projection and Rounding

In this section, we show how projection (w.r.t. Mahalanobis norm) can be done in time $O(n \log n)$ and rounding in time $O(n)$ for the uniform matroid. This projection algorithm also proves to be useful in case of partition matroid polytope. We also discuss a projection method on general matroid base polytopes, based on the method of Kumar and Bach [26], which needs to solve a total number of $n$ submodular function minimization (SFM) tasks (details below).

## D.1   Efficient projection on the uniform matroid

Let $G$ be a diagonal matrix with positive entries, $G = \text{diag}(g_1, \ldots, g_n)$. Our aim is to project a vector $\mathbf{y} \in \mathbb{R}_+^n$ on the uniform matroid base polytope defined as

$$P_k = \{\mathbf{x} \in \mathbb{R}_+^n \mid \sum x_i = k, \ 0 \leq x_i \leq 1\}.$$

The polytope $P_k$ is the convex hull of all the vectors that have precisely $k$ ones and $n - k$ zeros. Projecting $\mathbf{y}$ onto $P_k$ entails finding a point $\mathbf{x}$ in $P_k$, such that

$$\mathbf{x} = \underset{\mathbf{x} \in P_k}{\text{argmin}} \|\mathbf{x} - \mathbf{y}\|_G^2 := \underset{\mathbf{x} \in P_k}{\text{argmin}} (\mathbf{x} - \mathbf{y})^\top G (\mathbf{x} - \mathbf{y}),$$

where $\|\cdot\|_G$ is the Mahalanobis norm (i.e. the Mahalanobis distance to $\mathbf{0}$). Note that in the special case of $G = I$, this problem boils down to orthogonal projection of $\mathbf{y}$ onto $P_k$. We first transform this problem into an orthogonal projection, and solve that projection in $O(n \log n)$.

$$
\begin{aligned}
\mathbf{x} &= \underset{\mathbf{x} \in P_k}{\text{argmin}} (\mathbf{x} - \mathbf{y})^\top G (\mathbf{x} - \mathbf{y}) \\
&= \underset{\mathbf{x} \in P_k}{\text{argmin}} \|\mathbf{u} - \mathbf{w}\|_2^2, \quad \text{where } \mathbf{u} = G^{1/2}\mathbf{x} \text{ and } \mathbf{w} = G^{1/2}\mathbf{y} \\
&= G^{-1/2} \underset{\mathbf{u} \in G^{1/2} P_k}{\text{argmin}} \|\mathbf{u} - \mathbf{w}\|_2^2,
\end{aligned}
\tag{13}
$$

where (13) suggests an orthogonal projection on the polytope $G^{1/2} P_k$. By defining the vector $\mathbf{c} = (g_1^{-1/2}, \ldots, g_n^{-1/2})$, one has $G^{1/2} P_k = \{\mathbf{x} \in \mathbb{R}_+^n \mid \mathbf{c}^\top \mathbf{x} = k, 0 \leq x_i \leq \frac{1}{c_i}\}$. Theorem 6 shows that this projection can be done in $O(n \log n)$, and Algorithm 2 depicts the algorithm achieving the solution.

**Theorem 6.** *Let* $P = \{\mathbf{x} \in \mathbb{R}_+^n \mid \mathbf{c}^\top \mathbf{x} = k, 0 \leq x_i \leq \frac{1}{c_i}\}$, *where* $\mathbf{c} \in \mathbb{R}_+^n$ *is given. Then for any given point* $\mathbf{y} \in \mathbb{R}_+^n$ *one can find the solution to* $\text{argmin}_{\mathbf{x} \in P} \frac{1}{2}\|\mathbf{x} - \mathbf{y}\|_2^2$ *in* $O(n \log n)$ *time. Moreover this solution is unique.*

*Proof.* Let us begin by writing the KKT optimality conditions for the projected vector $\mathbf{x}$. The Lagrangian is defined by

$$\mathcal{L}(\mathbf{x}, \alpha, \boldsymbol{\beta}, \boldsymbol{\gamma}) = \frac{1}{2}\|\mathbf{x} - \mathbf{y}\|_2^2 + \alpha(\mathbf{c}^\top \mathbf{x} - k) - \boldsymbol{\beta}^\top \mathbf{x} + \boldsymbol{\gamma}^\top(\mathbf{x} - 1/\mathbf{c}),$$

---

**Algorithm 2** Projection on the Scaled Uniform Matroid Polytope

---

1: **Input:** vectors $\mathbf{y}, \mathbf{c} \in \mathbb{R}^n_+$ and $k \in \mathbb{N}$, s.t. $k \leq n$.
2: $\underline{\alpha}_i \leftarrow \frac{y_i c_i - 1}{c_i^2}, \bar{\alpha}_i \leftarrow \frac{y_i}{c_i}, \forall i \in [n]$
3: $S \leftarrow \{\underline{\alpha}_i\} \cup \{\bar{\alpha}_i\}$
4: Sort elements in $S$, so that $S = \{\alpha_1 < \ldots < \alpha_s\}$
5: $h \leftarrow n, \quad \alpha \leftarrow \min S - 1, \quad m \leftarrow 0$
6: **for** $i \in [s]$ **do**
7: $\quad h' \leftarrow h + (\alpha_i - \alpha)m$ {calculate function value at the new point using current slope $m$}
$\qquad$ {check if $\alpha^*$ is between $\alpha_i$ and $\alpha_{i-1}$}
8: $\quad$ **if** $h' < k \leq h$ **then**
9: $\qquad \alpha^* \leftarrow (\alpha_i - \alpha)\frac{h-k}{h-h'} + \alpha$
10: $\qquad$ return the projected vector $\mathbf{x}$ as follows:

$$
x_j = \left\{
\begin{array}{ll}
1/c_j & \alpha^* < \underline{\alpha}_j \\
y_j - \alpha^* c_j & \underline{\alpha}_j \leq \alpha^* \leq \bar{\alpha}_j \\
0 & \bar{\alpha}_j < \alpha^*
\end{array}
\right.
$$

11: $\quad$ **end if**
12: $\quad m \leftarrow m - \sum_{j:\underline{\alpha}_j = \alpha} c_j^2$ {for these $j$, $x(\alpha)_j$'s slope is changing from $0$ to $-c_j$}
13: $\quad m \leftarrow m + \sum_{j:\bar{\alpha}_j = \alpha} c_j^2$ {for these $j$, $x(\alpha)_j$'s slope is changing from $-c_j$ to $0$}
14: $\quad h \leftarrow h', \quad \alpha \leftarrow \alpha_i$
15: **end for**

---

where $\alpha \in \mathbb{R}$ and $\boldsymbol{\beta}, \boldsymbol{\gamma} \in \mathbb{R}^n_+$. Minimizing the Lagrangian w.r.t. $\mathbf{x}$ gives for each $i \in [n]$:

$$
x_i = y_i - \alpha c_i + \beta_i - \gamma_i, \tag{14}
$$

and also considering complementary slackness, we should have $\beta_i x_i = 0$ and $\gamma_i(x_i - 1/c_i) = 0$. If one provides suitable $\mathbf{x}$ and $\alpha, \boldsymbol{\beta}, \boldsymbol{\gamma}$ that satisfy the equations above, then $\mathbf{x}$ would be the optimal solution. In what follows, we construct $\mathbf{x}$ and provide suitable $\alpha, \boldsymbol{\beta}, \boldsymbol{\gamma}$.

For each $\alpha \in \mathbb{R}$, define $\mathbf{x}(\alpha) := \min\{\frac{1}{\mathbf{c}}, \max\{0, \mathbf{y} - \alpha\mathbf{c}\}\}$, where $\min$ and $\max$ are applied element-wise. By definition, one has $\mathbf{0} \leq \mathbf{x}(\alpha) \leq \frac{1}{\mathbf{c}}$. Let $h(\alpha) := \mathbf{c}^\top \mathbf{x}(\alpha)$. We claim that if for a value of $\alpha$, $h(\alpha) = k$, we are done, since $\mathbf{x}(\alpha) \in \tilde{P}$, and it satisfies the KKT conditions: If $x(\alpha)_i = 0$, by definition of $\mathbf{x}(\alpha)$ it means that $y_i - \alpha c_i \leq 0$, so we can set $\beta_i = -(y_i - \alpha c_i) \geq 0$ and $\gamma_i = 0$. If $x(\alpha)_i = \frac{1}{c_i}$, it means $y_i - \alpha c_i \geq \frac{1}{c_i}$, so we can set $\beta_i = 0$ and $\gamma_i = y_i - \alpha c_i - \frac{1}{c_i} \geq 0$. Otherwise, $0 < x(\alpha)_i < \frac{1}{c_i}$, which in that case we set $\beta_i = \gamma_i = 0$.

So it suffices to provide an $\alpha$ such that $h(\alpha) = k$. For each $i \in [n]$, define $\underline{\alpha}_i := \frac{y_i c_i - 1}{c_i^2}$ and $\bar{\alpha}_i := \frac{y_i}{c_i}$. It's obvious that if $\alpha \leq \underline{\alpha}_i$ then $x(\alpha)_i = \frac{1}{c_i}$, if $\alpha \geq \bar{\alpha}_i$ then $x(\alpha)_i = 0$, and otherwise $x(\alpha)_i = y_i - \alpha c_i$. So $x(\alpha)_i$ is a continuous decreasing function, and so will be $h(\alpha)$. Note that if $\alpha \leq \min\{\underline{\alpha}_i\}$, then $h(\alpha) = n$ and if $\alpha \geq \max\{\underline{\alpha}_i\}$, then $h(\alpha) = 0$. So by continuity, there is some $\alpha^*$ such that $h(\alpha^*) = k$. Now let $\alpha_1 < \ldots < \alpha_s$ be the set of all distinct values among $\underline{\alpha}_i$ and $\bar{\alpha}_i$. It's clear that for all $\alpha \in [\alpha_i, \alpha_{i+1}]$, $h(\cdot)$ is a linear function. By exploiting this fact, we can find $\alpha^*$ by searching through these endpoints. Detailed procedure is explained in Algorithm 2. $\qquad\square$

### D.2 Efficient projection on Partition matroid base polytope

Let $V$ be a ground set and $A_1, \ldots, A_m$ be a partition of $V$. A *partition matroid*, includes all sets $S \subseteq V$ such that for all $i \in [m]$ we have $|A_i \cap S| \leq k$. It's easy to see that the base polytope would be

$$
\mathcal{P} = \left\{ \mathbf{x} \in [0,1]^V \mid \forall i \in [m] : \sum_{j \in A_i} x_j = k \right\}.
$$

In order to project onto $\mathcal{P}$, we first note that it becomes a separable objective, partitioned over $A_i$. This means that it is sufficient to project $\mathbf{y}|_{A_i}$ onto the uniform matroid of $A_i$, for all $i \in [m]$. Since each projection takes $\mathcal{O}(|A_i| \log |A_i|)$ time, the total process would be $\mathcal{O}(n \log n)$.

### D.3 Projection on general matroid base polytopes

Let us now ask whether there is an efficient projection algorithm for general matroid polytopes. Here, we argue that the method proposed by Kumar and Bach [26] would be a reasonable candidate in the case of general matroid polytopes.

Let $g : 2^V \to \mathbb{R}_+$ be a submodular function, such that $g(\emptyset) = 0$, and let $g_L : \mathbb{R}^n \to \mathbb{R}_+$ be its Lovasz extension. We define the *base polytope* of $g$ as the set

$$\mathcal{B} = \{\mathbf{s} \in \mathbb{R}^n \mid \mathbf{s}(V) = g(V), \forall A \subset V : \mathbf{s}(A) \leq g(A)\}.$$

It can be shown [2] that the Lovasz extension is the *support function* of this polytope, i.e.

$$g_L(\mathbf{x}) = \sup_{\mathbf{s} \in \mathcal{B}} \mathbf{s}^\top \mathbf{x}. \tag{15}$$

For any $\mathbf{y} \in \mathbb{R}^n$ consider the task of minimizing the following objective with respect to $\mathbf{x} \in \mathbb{R}^n$:

$$g_L(\mathbf{x}) - \mathbf{y}^\top \mathbf{x} + \tfrac{1}{2}\|\mathbf{x}\|_2^2. \tag{16}$$

By using (15), we can rewrite (16) in the following dual form

$$\min_{\mathbf{x} \in \mathbb{R}^n} g_L(\mathbf{x}) - \mathbf{y}^\top \mathbf{x} + \tfrac{1}{2}\|\mathbf{x}\|_2^2 = \max_{\mathbf{s} \in \mathcal{B}} -\tfrac{1}{2}\|\mathbf{s} - \mathbf{y}\|_2^2, \tag{17}$$

in which the latter expression is precisely the projection of $\mathbf{y}$ on $\mathcal{B}$. In Kumar and Bach [26], the authors have exploited the structural properties of the Lovasz extension and the faces of the base polytope to create the so-called "Active-set" algorithm. The Active-set algorithm iteratively solves instances of isotonic regression as well as submodular function minimization tasks, whose overall complexity is less than a single submodular function minimization call (recall that by submodular function minimization, we mean the task of solving $\min_{\mathbf{x} \in [0,1]^n}(g_L(\mathbf{x}) - \mathbf{y}^\top \mathbf{x})$). By knowing (17), the algorithm can be viewed as a sequence of iterative projections on outer-approximations of the base polytope.

For any matroid, its associated rank function is a monotone submodular function. Also, the base polytope for a matroid's rank function is exactly the matroid base polytope. As a result of (17), we can use the Active-set algorithm to perform projections on the matroid base polytope. Interestingly, in the case of uniform matroids, the main parts of our projection scheme has similar counterparts as in the Active-set scheme. However, runtime complexity is significantly different due to several differences such as optimality checks: In our approach, this check is done in $\mathcal{O}(1)$, but in Active-set scheme, in each iteration, one should solve approximately $\mathcal{O}(n)$ submodular minimization tasks. However, the Active-set approach is more general, as explained above.

### D.4 The RANDOMIZED-PIPAGE-ROUNDING procedure

The randomized pipage rounding procedure was first proposed in [7] for any matroid $\mathcal{M}$. Here, we show how this procedure can be efficiently done (in linear time) for the uniform matroid. Suppose we have a matroid $\mathcal{M}$ and a point $\mathbf{y} := (y_1, \cdots, y_n)$ in its corresponding base polytope. We want to round $\mathbf{y}$ to a vertex of the base polytope. In each step of the algorithm, one has a fractional solution $\mathbf{y}$ and a tight set $T$ containing at least two fractional variables (recall that if the matroid rank function is $r(\cdot)$, a set $T$ is tight if $\mathbf{y}(T) = r(T)$; Tight sets are exactly those constraints in the base polytope who are tight at $y$). It modifies two fractional variables in such a way that their sum remains constant, until some variable becomes integral or a new constraint becomes tight. Note that since the sum of all of elements of $\mathbf{y}$ is an integer (rank of the matroid), there exist at least two fractional variables in the case that the point is fractional.

For our purpose, we are faced with uniform matroid, which we argue that finding tight constraints is easy, i.e., we can compute the HITCONSTRAINT subroutine in a very fast way. This subroutine is given a fractional point $\mathbf{y}$ and two variables $i$ and $j$, and tries to increase $y_i$ and decrease $y_j$ simultaneously, and find a new tight constraint $A$. For sure, one should search for this new tight set through the sets having $i$ inside them but not $j$. So let $\mathcal{A}$ denote the family of all subsets containing $i$ and not containing $j$. So we are interested in $\delta = \min_{A \in \mathcal{A}}(r(A) - \mathbf{y}(A))$, the maximum increase in $y_i$ (and decrease in $y_j$) that does not violate any polytope condition, but produces a new tight constraint. We claim that $\delta$ is trivial in case of the uniform matroid: $\delta = \min\{1 - y_i, y_j\}$. Also the new tight set $A$ is either $\{i\}$ or $V - j$.

This simple form of the HITCONSTRAINT gives an efficient algorithm for RANDOMIZED-PIPAGE-ROUNDING, which we describe in Algorithm 3. Moreover, one has the following Theorem:

---

**Algorithm 3** RANDOMIZED-PIPAGE-ROUNDING for the Uniform Matroid

---

1: **Input:** fractional $\mathbf{y}$; $k \in \mathbb{N}$ defining the matroid rank
2: **while** $\mathbf{y}$ fractional **do**
3:     Select $i$ and $j$ among fractional variables
4:     **if** $y_i + y_j < 1$ **then**
5:         Let $p = y_j/(y_i + y_j)$
6:         With probability $p$, set $y_i \leftarrow 0$ and $y_j \leftarrow y_i + y_j$, and with probability $1 - p$, set $y_i \leftarrow y_i + y_j$
        and $y_j \leftarrow 0$.
7:     **else**
8:         Let $p = (1 - y_i)/(2 - y_i - y_j)$
9:         With probability $p$, set $y_i \leftarrow y_i + y_j - 1$ and $y_j \leftarrow 1$, and with probability $1 - p$, set $y_i \leftarrow 1$
        and $y_j \leftarrow y_i + y_j - 1$.
10:    **end if**
11: **end while**
12: **return** $\mathbf{y}$

---

**Theorem 7.** *Let $\mathcal{M}$ be the uniform matroid and $\mathbf{y}$ be a fractional point inside $\mathcal{P}(\mathcal{M})$. Then* RANDOMIZED-PIPAGE-ROUNDING *returns an integral point $\mathbf{y}_{rnd}$ in $\mathcal{O}(n)$ time, such that*

$$\mathbb{E}[F(\mathbf{y}_{rnd})] \geq F(\mathbf{y}).$$

The proof of this algorithm's correctness is similar to the original one given in [7]. It is also noteworthy that our algorithm runs in $\mathcal{O}(n)$ time compared to $\mathcal{O}(n^2)$, as described in [7].

## E   Details on experiments

### E.1   Influence Maximization

Our approach is to obtain samples from the product distribution, $(G, v) \sim \mathcal{G} \times \mathcal{U}$, and compute the set $P_v$ (see the definitions for the class of weighted coverage functions in Section 3). Note that the vertex $v$ is chosen uniformly at random. Since $P_v$ is smaller compared to $G$, it is less efficient to sample $G$ completely. Instead, while doing the BFS starting from $v$, we select edges with probability $p$ and proceed. Note that in case of maximizing the upper-bound (8), whenever the sum of $x_i$ visited so far exceeds 1, one can stop and return $\mathbf{0}$, otherwise return $\mathbf{1}_{P_v}$ in the end. This approach is quite fast, but may need too many iterations to converge, because of its locality (i.e. we only take one vertex in each iteration). Note that the size of the gradient in this case is at most $\sqrt{|P_v|} \leq \sqrt{n}$.

### E.2   Facility Location

Computing the (stochastic) gradient for the concave upper bound can be done in linear time. Let $h$ be the first index that $\sum_{j=1}^{h} x_j \geq 1$, then a vector in sub-gradient of $\bar{F}(\cdot)$ is simply

$$\mathbf{g} = (m_1 - m_h, \ldots, m_{h-1} - m_h, 0, 0, \ldots, 0). \tag{18}$$

In case of the multilinear extension, we give a linear time algorithm for computing the gradient. Let $h$ be the first index that $x_h = 1$ (if no such index exists, then set $h = n + 1$). It's clear from (10) that $\frac{\partial F}{\partial x_i}(\mathbf{x}) = 0$ for $i = h + 1, \ldots, n$. For $i = h, h - 1, \ldots, 1$ one has the following recursion:

$$\frac{\partial F_e}{\partial x_i}(\mathbf{x}) = \frac{1 - x_{i+1}}{1 - x_i} \frac{\partial F_e}{\partial x_{i+1}}(\mathbf{x}) + (m_i - m_{i+1}) \prod_{j=1}^{i-1} (1 - x_j),$$

which can be done completely in linear time.

### E.3   Exemplar-based Clustering

Let $V$ be a set of points. One can quantify the representativeness a set of exemplars $S \subseteq V$ by the loss function $L(S) = \frac{1}{|S|} \sum_{v \in V} \min_{s \in S} \|v - s\|_2$. Finding the best $k$ exemplars is equivalent to

solving $\min_{|S|=k} L(S)$. By introducing an appropriate phantom element $e_0$ we can turn $L(\cdot)$ into a monotone submodular function [15]: $f(S) = L(\{e_0\}) - L(S \cup \{e_0\})$. Thus maximizing $f$ is equivalent to minimizing $L$. In our experiments, to ensure non-negativity of the function values, we transform our dataset $V$ by the transformation $T : \mathbb{R}^m \to \mathbb{R}^m$,

$$T(\mathbf{x}) = \frac{3}{\sqrt{m}}\mathbf{1} + \frac{\mathbf{x} - \bar{\mathbf{x}}}{\|\mathbf{x} - \bar{\mathbf{x}}\|_2}, \quad \text{where } \bar{\mathbf{x}} = \frac{1}{|V|}\sum_{\mathbf{x} \in V} \mathbf{x},$$

and set $e_0 = \mathbf{0}$.

# F  Concave Envelope Evaluation

Here we prove Lemma 1. Define $f(\mathbf{x})$ as follows:

$$f(\mathbf{x}) = \begin{cases} 1 - \prod_{i=1}^n (1 - x_i) & \mathbf{x} \in [0, 1]^n \\ -\infty & \text{Otherwise} \end{cases}.$$

We first compute the Fenchel concave dual of $f$, which is defined as

$$f_*(\mathbf{y}) = \inf\{\mathbf{y}^\top \mathbf{x} - f(\mathbf{x}) \mid \mathbf{x} \in \mathbb{R}^n\}. \tag{19}$$

For brevity, let us define $h(\mathbf{x}, \mathbf{y}) := \mathbf{y}^\top \mathbf{x} - f(\mathbf{x})$. We partition $\mathbb{R}^n$ into several subsets (cases below) and compute the infimum (19) for each subset and take the minimum over all partitions.

Case I, $\mathbf{x} \in (0, 1)^n$: Here we can compute the infimum by setting the gradient equal to zero. For a fixed $\mathbf{y} \in \mathbb{R}^n$ we have

$$\nabla_{\mathbf{x}} h(\mathbf{x}, \mathbf{y}) = \mathbf{0} \iff y_i = \frac{\partial f}{\partial x_i} = \frac{\prod_{j=1}^n (1 - x_j)}{(1 - x_i)}.$$

Clearly $y_i > 0$. Let us define $P = \prod_i (1 - x_i)$. We then have $y_1 \cdots y_n = P^{n-1}$, and then $x_i = 1 - P/y_i$. Since $x_i > 0$, we should have $y_i > P$. The following lemma gives the necessary condition on $\mathbf{y}$ for this to happen.

**Lemma 8.** Let $y_1, \ldots, y_n \in \mathbb{R}^+$, $n \geq 2$, and assume that $\forall i \in [n] : y_i > \sqrt[n-1]{y_1 \cdots y_n}$. Then we have $y_i < 1$ for all $i \in [n]$.

*Proof.* We prove the argument by induction. The case $n = 2$ is obvious since

$$y_1 > y_1 y_2 \Rightarrow y_2 < 1, \quad y_2 > y_1 y_2 \Rightarrow y_1 < 1.$$

Suppose the claim is true for $n - 1$. We now prove it for $n$. W.l.o.g. assume $y_1 \leq \cdots \leq y_n$. We have

$$y_1^{n-1} > y_1 y_2 \cdots y_n \Rightarrow \forall i \geq 2 : y_i \geq y_1 > \sqrt[n-2]{y_2 \cdots y_n}.$$

So $y_2, \ldots, y_n$ satisfy the lemma's conditions, and by the induction hypothesis, we have $y_2, \ldots, y_n < 1$. Since $y_1 \leq y_2$, we also have $y_1 < 1$. $\square$

So far, we know that there is a minimum in this case if $\mathbf{y} \in (0, 1)^n$. The minimum value would be

$$h(\mathbf{x}, \mathbf{y}) = \sum x_i y_i - f(\mathbf{x}) = \sum (1 - P/y_i) y_i - (1 - P)$$

$$= \sum y_i - (n-1)P - 1 \tag{20}$$

This minimum value, as will be clear shortly, is not the best (lower values are available on other partitions), because of the following lemma:

**Lemma 9.** Let $\mathbf{y} \in (0, 1)^n$. Then the minimum value of $h(\mathbf{x}, \mathbf{y})$ over $\mathbf{x} \in (0, 1)^n$ is strictly greater than $-1$.

*Proof.* Because of (20), we have

$$\min_{\mathbf{x} \in (0,1)^n} h(\mathbf{x}, \mathbf{y}) = \sum_{i=1}^n y_i - (n-1)\sqrt[n-1]{y_1 \cdots y_n} - 1$$

$$> \sum_{i=2}^n y_i - (n-1)\sqrt[n-1]{y_2 \cdots y_n} - 1 \qquad \text{since } 1 > y_1 > 0$$

$$\geq -1 \qquad\qquad \text{by AM-GM inequality}$$

$\square$

Case II, at least for one index $i \in [n]$ we have $x_i = 1$: In this case, we have $f(\mathbf{x}) = 1$, and $h(\mathbf{x}, \mathbf{y}) = \mathbf{y}^\top \mathbf{x} - 1$. W.l.o.g. assume $y_1 \leq \cdots \leq y_n$. It's clear that the minimum value for $h$ over these values of $\mathbf{x}$ would be

$$\min_{\mathbf{x} \in [0,1]^n, \exists_i x_i = 1} h(\mathbf{x}, \mathbf{y}) = \begin{cases} y_1 - 1 & \mathbf{y} \geq \mathbf{0} \\ \sum_{y_i < 0} y_i - 1 & \text{Otherwise} \end{cases}.$$

Case III, for some $i \in [n]$ we have $x_i = 0$: In this case, $f$ would have the same form (with $x_i$ deleted) and also $y_i$ is deleted from $h$, and hence the problem is reduced to $n - 1$ dimensional case. So the same type of solutions as the previous cases would occur.

In total, for $\mathbf{y}$, such that $y_1 \leq \cdots y_n$ we can write

$$f_*(\mathbf{y}) = \begin{cases} \sum_{y_i < 0} y_i - 1 & y_1 < 0 \\ y_1 - 1 & 0 \leq y_1 < 1 \\ 0 & 1 \leq y_1 \end{cases}$$

Now that we have computed the Fenchel dual of $f$, we can compute the Fenchel dual of $f_*$, which would be the Fenchel bi-conjugate of $f$. By definition we have

$$f_{**}(\mathbf{z}) = \inf\{\mathbf{z}^\top \mathbf{y} - f_*(\mathbf{y}) \mid \mathbf{y} \in \mathbb{R}^n\}.$$

Let's define $g(\mathbf{y}, \mathbf{z}) = \mathbf{z}^\top \mathbf{y} - f_*(\mathbf{y})$. We will find the minimum on the orthant $y_1 \leq \cdots \leq y_n$.

Case IV, $y_1 \geq 1$: We have $g(\mathbf{y}, \mathbf{z}) = \mathbf{z}^\top \mathbf{y}$. If $\mathbf{z}$ has negative components, then infimum would become $-\infty$. So from now on, we assume $\mathbf{z} \geq 0$. In this case, the minimum of $g$ over this case is $\sum z_i$.

Case V, $y_i \in (0, 1)$: We have

$$g(\mathbf{y}, \mathbf{z}) = \mathbf{z}^\top \mathbf{y} - y_1 + 1 \geq y_1(\sum z_i - 1) + 1.$$

Now if $\sum z_i \geq 1$, the righthand side's minimum would be 1 and this minimum is achieved by setting $\mathbf{y} = 0$. But if $\sum z_i < 1$, then

$$y_1(\sum z_i - 1) + 1 \geq \sum z_i - 1 + 1 = \sum z_i,$$

and this minimum is achieved by setting $\mathbf{y} = \mathbf{1}$.

Case VI, $y_1 < 0$: We have

$$g(\mathbf{y}, \mathbf{z}) = \mathbf{z}^\top \mathbf{y} - \sum_{y_i < 0} y_i + 1$$

$$= \sum_{i:y_i < 0} y_i(z_i - 1) + \sum_{j:y_j \geq 0} y_j z_j + 1$$

If for some $i$, $z_i > 1$, then infimum of $g$ over this case would become $-\infty$, so we have also $\mathbf{z} \leq \mathbf{1}$. In this case, the minimum would become 1.

Summing all up, we have:

$$f_{**}(\mathbf{z}) = \begin{cases} \min\{1, \sum z_i\} & \mathbf{z} \in [0, 1]^n \\ -\infty & \text{Otherwise} \end{cases},$$

and this is what we wanted to prove.

# G   Pathological Examples

Here we present a special case, where GREEDY fails to give a proper solution, but our method works well. Our example would be about influence maximization with partition matroid condition. It is well known that for general matroids (matroids other than the uniform matroid), GREEDY is guaranteed to give $1/2$-optimal solution.

Figure 2: Pathological graph for Influence Maximization

Construct a graph $G = (V, E)$ as follows: $V = \{0, 1, 2, \ldots, 2N + 1\}$, and connect vertices like in the figure. Also take the partitions to be $\{0\}$ and $\{1, 2, \ldots, 2N + 1\}$, meaning that one should take from each partition at most one vertex.

The GREEDY algorithm, first chooses the vertex with the highest out-degree, which is $1$, and then is forced to choose $0$ as the second vertex because of matroid condition, leading to the $(1/2 + \epsilon)$-optimal answer $\{0, 1\}$.

On the other hand, our algorithm successfully gives the optimal answer $\{0, 2\}$. It's a good practice to show why. Let us choose $\mathbf{x}^{(0)}$ be the projection of $\mathbf{1}$ on the partition matroid polytope, namely

$$\mathbf{x}^{(0)} = (1, \tfrac{1}{2N+1}, \ldots, \tfrac{1}{2N+1}).$$

The (sub-)gradient of $\bar{F}$ at $\mathbf{x}^{(0)}$ is calculated as follows:

$$\nabla \bar{F}(\mathbf{x}^{(0)}) = \mathbf{0}_{\{0\}} + \mathbf{1}_{\{1\}} + \mathbf{1}_{\{2\}} + \mathbf{1}_{\{3,1\}} + \sum_{a \in A} \mathbf{0}_{\{0,1,a\}} + \sum_{b \in B} \mathbf{1}_{\{2,b\}}$$

$$= (0, 2, N, 1, \underbrace{0, \ldots, 0}_{a \in A}, \underbrace{1, \ldots, 1}_{b \in B})$$

where the reason of $\mathbf{0}_{\{0\}}$ and $\mathbf{0}_{\{0,1,a\}}$ for $a \in A$ is $\mathbf{x}_0 = 1$ and $\mathbf{x}_0 + \mathbf{x}_1 + \mathbf{x}_a > 1$ respectively. It's obvious that moving along this gradient, and projecting, makes $\mathbf{x}_0 = \mathbf{x}_2 = 1$ and all others to zero, selecting $\{0, 2\}$ as the solution.

The difference here with GREEDY is apparently in two places: (i) being on the matroid base polytope forces the algorithm tochoose a vertex from the first partition, and simultaneously (ii) selecting $0$ means all vertices $a \in A$ are influenced, so there is no need to select $1$. GREEDY does not take into account that in the future it should select a node in some other partition, that may lose his achievement in the first step.