[Reviews · NeurIPS 2017]

Reviewer 1



The papers deals with the problem of submodular maximization; specifically, it proposes a stochastic optimization algorithm for maximizing a specific family of submodular functions, i.e. weighted coverage, under matroid constraints. The algorithm operates on the multilinear extension of the weighted coverage function. This way, the authors are sacrificing accuracy by optimizing a concave function which is a bounded approximation of the target function. However, they gain theoretically supported bounds on the convergence rate and the running time, which they also showcase in the experiments. In general, the paper is well written, and the set of ideas that it uses are well put together. The experimental section, although brief, drives the point that the authors want to make. From the application point of view, the influence maximization is quite strong and interesting, with potential impact to the practitioners as well. This is not the case for the facility location though, which feels very forced and lacks in presentation. In total, the techinical novelty of the paper is very limited, as it mostly brings together ideas from the literature. Although it is an interesting paper to read, it does not have neither a groundbreaking idea, a groundbreaking application nor a groundbreaking theory. More detailed comments: line 8: rewrite the sentence around "formalization" line 146: the notation B_v is a little strange. Maybe introduce V as {B_1, ... B_|V|} ? line 190: \bar{x}^* = \argmax and not \in line 199: F^* -> \barF^* line 208: denote -> denotes line 211: form -> from line 241: how is the facility location problem related to the examplar-based clustering that is being used before and after this section? Why not use one name instead? line 241+: This section comes out as over-complicated. It is hard to parse and it requires going back and forth many times. Could be either because of notation or because it lacks an intuitive explanation. line 260: two two problem classes line 264+ : The use of apostrophe as the thousands sign is a bit strange to get used to. Maybe leave a small space between the numbers instead? line 267: Not sure what the last sentence means. line 270: In the Table and the Figure, the term LAZYGREEDY is being used (together with a number in the Figure). What is the difference between GREEDY and LAZYGREEDY? What does the number represent? caption of Table 1: There is an orphan footnote.

Reviewer 2



This paper considers the problem of maximizing a weighted coverage function subject to matroid constraints in the stochastic setting. For weighted coverage functions, the multilinear extension, which is typically optimized instead of the discrete objective, is known to admit a concave upper bound which is at most a factor of (1 - 1/e) away from it. The authors propose to optimize the concave upper bound via stochastic project gradient descent, thus obtaining a stochastic algorithm which returns a (1-1/e)-approximation to the original problem, after applying pipage rounding. I couldn't identify any reasonable contribution of this work. By restricting the problem considered to the case of weighted coverage function, the solution presented is quite straightforward: For this particular class of functions, it is known that the multilinear extension can be approximated by a concave function which is at most a factor of (1 - 1/e) away from it. Thus stochastic projected gradient descent can be of course used to maximize the concave approximation, since now it's a concave maximization problem. Besides this nice but straightforward observation, the only other novel result is Proposition 3 which bounds the number of samples needed by the greedy algorithm, when ran over the emipirical objective function subject to cardinality constraint. As noted in the proof, this proposition follows directly from Hoeffding's inequality and the union bound. The paper is well written but does not present any substantial contribution to warrant a nips publication.

Reviewer 3



The paper discusses the development of stochastic optimization for submodular coverage functions. The results and proofs are a mish-mash of existing results and techniques; nevertheless the results are novel and should spark interest for discussions and further research directions. The central contribution of the paper is providing with a framework to exploit the SGD machinery. To this effect, coverage functions can be harnessed, because the concave upperbound is obtained easily. While this seems restrictive, the authors present applications with good empirical performance vs classic greedy. I have a couple of points:- (a) Is there to use projected SGD over Stochastic Frank-Wolfe [A] ? The latter will get rid of messy projection issues for general matroids, albeit the convergence might be slower. (b) An interesting aspect of the paper is connection between discrete optimization and convex optimization. I would suggest including more related work that discuss these connections [B,C,D,E]. [B] also talks about approx. form of submodularity, [C,D] both follow the notion of “lifting” atomic optimization to continuous domain. Minor: The paper is well-written overall. There are minor typos for which I suggest the authors spend some time cleaning up the paper. E.g. Line 22,24 “trough”, line 373 “mosst”. An equation reference is missing in the appendix. Numbering of Proof Proposition 2 is messed up. The proof of Theorem 2 seems to be missing, I understand it follows from pipage rounding, but it might be a good idea to give a Lemma or something with reference from which approx. guarantee for rounding follows. In general, adding more information about pipage rounding in the appendix for completeness might be a good idea. The main text is missing a page of references. [A] Linear Convergence of Stochastic Frank Wolfe Variants. Donald Goldfarb, Garud Iyengar, Chaoxu Zhou [B] Restricted Strong Convexity Implies Weak Submodularity. Ethan R. Elenberg , Rajiv Khanna , Alexandros G. Dimakis , and Sahand Negahban [C] On Approximation Guarantees for Greedy Low Rank Optimization Rajiv Khanna , Ethan R. Elenberg1 , Alexandros G. Dimakis , and Sahand Negahban [D] Lifted coordinate descent for learning with trace-norm regularization. Miro Dudik Zaid Harchaoui , Jérôme Malick [E] Guaranteed Non-convex Optimization: Submodular Maximization over Continuous Domains. Andrew An Bian, Baharan Mirzasoleiman, Joachim M. Buhmann, Andreas Krause